# Long term declines in the functional diversity of sharks in the coastal oceans of eastern Australia
Christopher J. Henderson [1] ✉, Ben L. Gilby[2], Mischa P. Turschwell[3], Lucy A. Goodridge Gaines[1], Jesse D. Mosman[1], Thomas A. Schlacher[1], Hayden P. Borland[1] & Andrew D. Olds[1]

Human impacts lead to widespread changes in the abundance, diversity and traits of shark assemblages, altering the functioning of coastal ecosystems. The functional consequences of shark declines are often poorly understood due to the absence of empirical data describing long-term change. We use data from the Queensland Shark Control Program in eastern Australia, which has deployed mesh nets and baited hooks across 80 beaches using standardised methodologies since 1962. We illustrate consistent declines in shark functional richness quantified using both ecological (e.g., feeding, habitat and movement) and morphological (e.g., size, morphology) traits, and this corresponds with declining ecological functioning. We demonstrate a community shift from targeted apex sharks to a greater functional richness of non-target species. Declines in apex shark functional richness and corresponding changes in non-target species may lead to an anthropogenically induced trophic cascade. We suggest that repairing diminished shark populations is crucial for the stability of coastal ecosystems.

Anthropogenic impacts to natural landscapes and food webs are ubiquitous and have resulted in fundamental changes to ecosystem structure, functioning and key ecosystem services globally[1–3]. Species loss results in declines in the key ecological functions that maintain the condition and resilience of ecosystems[4,5]. Top predators have experienced significant declines globally due to anthropogenic activities[6–8]. Their loss can modify the population dynamics of prey species and the structure of food webs, and has implications for habitat forming species (e.g., corals or kelp), resulting in concurrent declines in key ecosystem services and ecological functions[9].

Sharks are functionally important components of coastal and oceanic food webs as they can exert top-down pressure on food webs at large spatial scales through the direct and indirect effects of predation[10,11]. They are also well researched, meaning that their abundance and diversity is well understood in many settings[8,12]. However, they are experiencing significant threats and declining due to human activities (e.g., overharvesting, climate change, habitat loss)[8,12–16], with overfishing alone driving one third of shark and ray species to extinction[17]. The removal of high order species such as sharks from marine food webs is amplified when the functional traits of these species are also lost from the community. Maintaining a diversity of traits within an ecosystem is crucial to ensure the continued provision of multiple key ecological functions across the food web[9,11]. As sharks have

evolved to become highly distinct[11], their losses can also lead to a simplification and contraction of functional trait space through reduced trait diversity, leading to the loss of ecological niches from food webs[14]. The functional consequences of sharks loss are, however, rarely described because these impacts often take decades to detect, and most long-term studies only highlight changes in the abundance of these species rather than broader functional effects[6,15,18].

The rate (e.g., the frequency or quantity) and distribution (e.g., the spatial scale) of ecological functions across landscapes are intrinsically linked to biodiversity because a greater variety of species performing a particular function will usually increase both the rate and stability of that function[3,4,19,20]. It is, however, not only the number of different species that controls the expression of an ecological function, but also the diversity of traits that these species possess that alters the intensity, magnitude and/or spatial distribution of ecological functions[21,22]. Traits are useful as they quantify the ecology of an organism by differentiating the role a species plays in an ecosystem, such as how and where they feed or the size of an individual species[22,23]. Due to high taxonomic diversity and evolutionary distinctiveness between shark species, understanding the functional composition of shark communities is of high interest[11]. Furthermore, understanding the functional traits of coastal predators such as sharks is crucial for coastal

---

[1]School of Science, Technology and Engineering, University of the Sunshine Coast, Maroochydore, QLD 4558, Australia. [2]School of Science, Technology and Engineering, University of the Sunshine Coast, Petrie, QLD 4558, Australia. [3]Coastal and Marine Research Centre, Australian Rivers Institute, Griffith University, Nathan, QLD 4111, Australia. ✉e-mail: chender1@usc.edu.au

management as sharks play a key role in the coastal food web and their loss is likely to have substantial consequences on the structure and functioning of that ecosystem[11,13,14,18]. Changes in functional diversity, a metric that quantifies variation in the set of traits possessed by all species in a community, is often linked to changes in the provision of multiple ecological functions linked to resilience in an ecosystem[20,21,24,25]. Functional diversity has been shown, therefore, to be a better predictor of ecosystem functioning and resilience than most conventional measures of diversity and so is increasingly used to assess how disturbance modifies ecological functioning[20,21,26].

We use data from the fisheries-independent Queensland Shark Control Program (QSCP), which has deployed a combination of mesh nets and baited hooks along 1760 km of coastline continuously since 1962[15]. Whilst the primary intent of the program is to remove large shark species (hereafter referred to as targeted apex shark species) thought to pose risks to swimmers, approximately 75% of species caught in the program are incidental catches. Because the field implementation of the program is standardised, and it incidentally captures many non-target species, it effectively represents a high density, high frequency, and long-term sampling of higher-order coastal fishes. It therefore offers a long-term record of the continual capture of sharks, rays and other teleosts along the Queensland coastline with a known effort[15]. We used this publicly available long-term catch data to demonstrate fundamental changes in the functional diversity, ecological functioning and species composition of targeted and non-targeted coastal species over six decades. Traditional measures which highlight changes in the abundance and diversity of species (e.g., species richness) may not be effective in quantifying the loss of functional roles in ecosystems[21,27,28]. Trait-based approaches are considered the most appropriate when information on ecological functioning is not completely available and so are widely used to assess ecosystem functioning change in ecological and evolutionary studies[21,22]. We characterise the functional traits of targeted apex sharks and non-targeted coastal fish species using two different approaches to calculate functional diversity. These two approaches included two separate types of trait values; one quantifying change in ecological traits (e.g., feeding group, habitat preference and movement scale) and the other quantifying change in

morphological traits (e.g., maximum total length, eye diameter, teeth morphology)[29,30]. We used these two approaches because we wanted to understand the effects of the shark control program from multiple functional perspectives. We hypothesise that widespread reductions in the numbers of targeted apex shark species will result in significant declines of functional diversity and ecological functioning over time[8,15].

## Results and discussion
### Declines in functional richness
We report significant and consistent declines in the functional diversity of targeted apex shark communities that correspond to substantial shifts in the overall shark community composition in eastern Australia over six decades. Bayesian generalised additive mixed models show that there was a 99.88% and 100% probability of a decline in the functional richness of targeted sharks, based on ecological and morphological traits respectively, over the last six decades (Fig. 1A). Sharp declines in functional richness, which calculates the total area of trait space, are associated with a shift towards a more compressed trait space over time (Fig. 1A). Nets captured a greater functional richness of targeted apex shark species, suggesting that most functional declines associated with the QSCP have been caused by the use of mesh nets (Fig. S1). Bayesian generalised additive mixed models show that there was a 100% and 99.84% probability of an increase in the functional richness of non-target coastal fish species, based on ecological and morphological traits respectively, during the last six decades (Fig. 1A). Increases in the functional richness of non-target species is illustrated by an overall increase in the functional variety of the species caught in the program over time. Mesh nets captured a greater functional richness, suggesting that the continued use of mesh nets in the QSCP continue to have a significantly detrimental impact on the assemblage of non-target coastal fish species (Fig. S1).

Prior to 1997, poorer identification classifications resulted in a higher number of individuals identified to genus level (e.g., 'whaler'). To account for this, we averaged the functional traits of all genus or groups within which an individual was identified because we wanted to limit the effects of new species being identified in the community after this time (e.g., those that

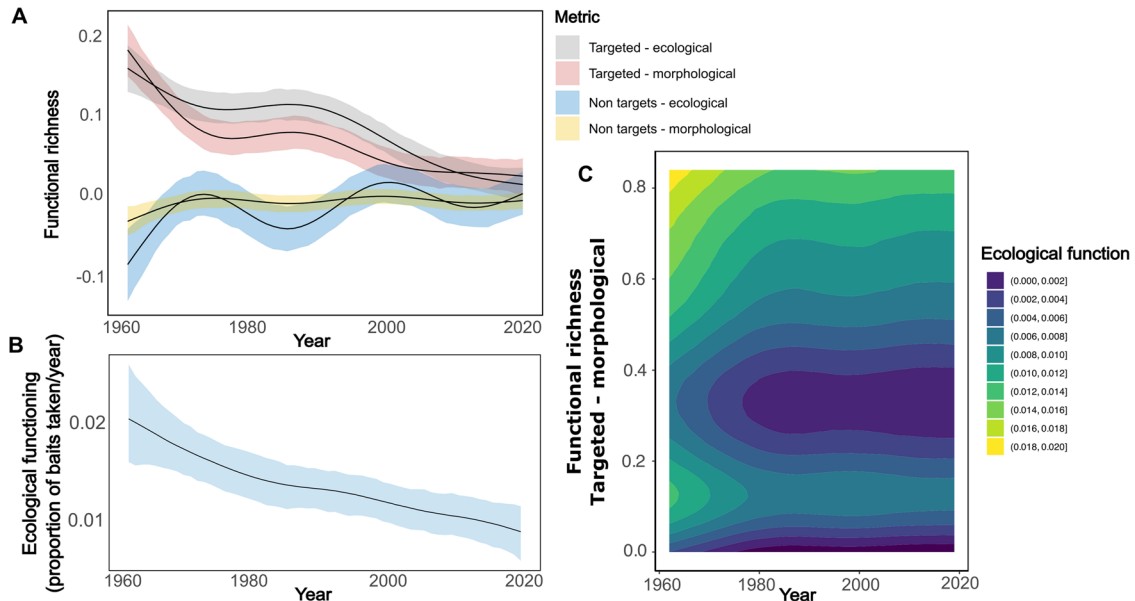

**Fig. 1 | Declines in functional richness over time.** The functional richness of targeted sharks decreased dramatically from 1962 to 2019. **A** These declines manifest as striking reductions in the functional richness of targeted sharks based on morphological and ecological species traits, and are followed by increases in the functional richness of non-target species based on morphological and ecological species traits. **B** These overall changes in functional richness manifest themselves in a shift in

the ecological functioning of coastal ecosystems. **C** Finally, ecological functioning, which is represented by the rate at which drumline baits are removed is significantly correlated with functional richness of targeted species based on morphological traits. Grey intervals represent the 95% credible intervals for the Bayesian GAMM and *n* = 5336 independent samples.

**Fig. 2 | Significant shifts in community composition over time.** A principle coordinates analysis plot highlighting the changes in the composition of targeted species in the Queensland Shark Control Program from 1962 to 2019. The change in composition is significant for both drumlines and nets, and targeted and non-targeted species with a clear shift in each community. Points in each plot represent centroids for each year, with the colour of that point going from dark blue (1962) to light blue (2019). Each point represents all beaches across all regions per year and is based on *n* = 5336 independent samples.

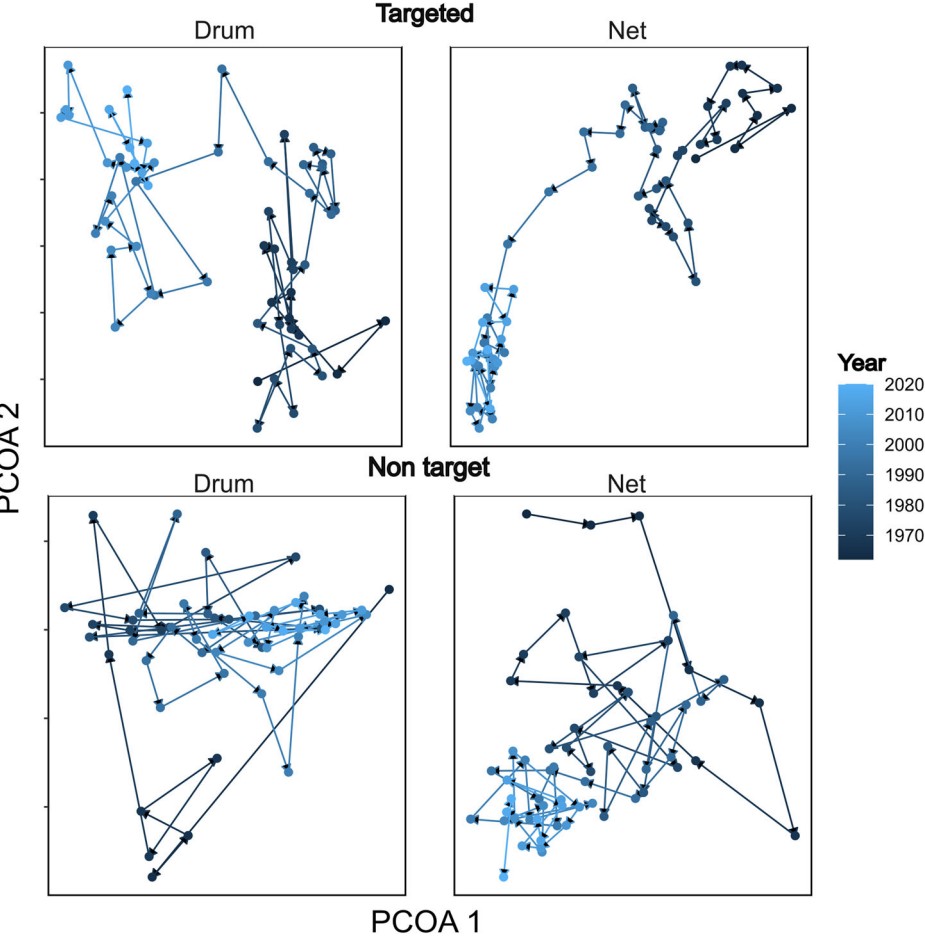

were previously mis-identified), even though the species was likely to have been caught previously. Therefore, we analysed the same effects on all targeted apex sharks without them being grouped to show that these patterns hold true (See Fig. S2). Crucially, the catch effort at and between individual beaches has not been consistent over the life of the program. Effort was included in all models assessing change in functional richness, with Bayesian generalized additive mixed models on functional richness there was always a greater than 90% chance of an effect of effort on functional richness. The effect of effort on the functional richness of targeted and non-targeted species was typically non-linear using both ecological and morphological traits, with functional richness typically reaching its maximum at low levels of effort and then maintaining that level as effort increased (See Fig. S3).

## Declines in ecological functioning

Bayesian generalised additive mixed models showed that there was a 99% probability that carrion consumption rates (i.e., the proportion of captures per drumline per year) declined significantly over six decades (Fig. 1B). We used proportion of captures as information on the removal of baits alone is not included in the public database and would be different between individual contractors. While this may be a crude metric of actual carrion consumption, as the full information on bait replacement is not kept, this is the most accurate, but likely still an underestimate, measure of function that can be identified in the current database. Carrion consumption was highest at the beginning of the program, and Bayesian generalised additive mixed models show that there was a 95.84% chance of a decline in functional richness metrics calculated based on morphological traits with declining carrion consumption rates (Fig. 1C). Here, we show that ecological functioning and functional richness were highest at the beginning of the program. Functioning, here carrion consumption, remains high when

functional richness is high, but this slightly decreased as time continued (Fig. 1C).

## Changes in species composition and traits over time

Significant declines in the functional diversity of targeted apex sharks (Fig. 1A) and increases for non-targeted species illustrate a significant shift in coastal ecosystem structure over six decades (Fig. 2). This has resulted in a change in program catch composition and declines in the catch of several large, iconic and threatened species (Fig. 2). This is evident for the catch composition of both targeted apex shark and non-targeted species in both nets and drumlines. Changes in the composition of catch in the QSCP is illustrated by an overall decline in the abundance of target and non-target species in the program (Fig. 3A) and significant declines in the average length of target and non-target species caught in the program. However, this effect was more pronounced for non-target species (Fig. 3B). Significant changes in the abundance of species caught in the program and changes in length is further highlighted by falling catches of the three main target species in the program, the great white (*Carcharodon carcharias*) (Fig. 3C), whaler sharks (*Carchahinidae* spp.) (Fig. 3D), and tiger sharks (*Galeocerdo cuvier*) (Fig. 3E).

## Shifts in the functional uniqueness and specialisation of species

Functional uniqueness (which identifies the overall isolation of a species and is linked to functional redundancy) and specialisation (which identifies species that contribute the most to functional richness) metrics are suitable for assessing the levels of redundancy in the community and whether species in the community are more generalist or specialist in nature. We found that targeted species that had higher uniqueness and specialisation (Fig. 4A, B, Table S2) and were typically more endangered, while non-targeted species were from a variety of threatened categories (Fig. 4C, D, Table S2). Bayesian

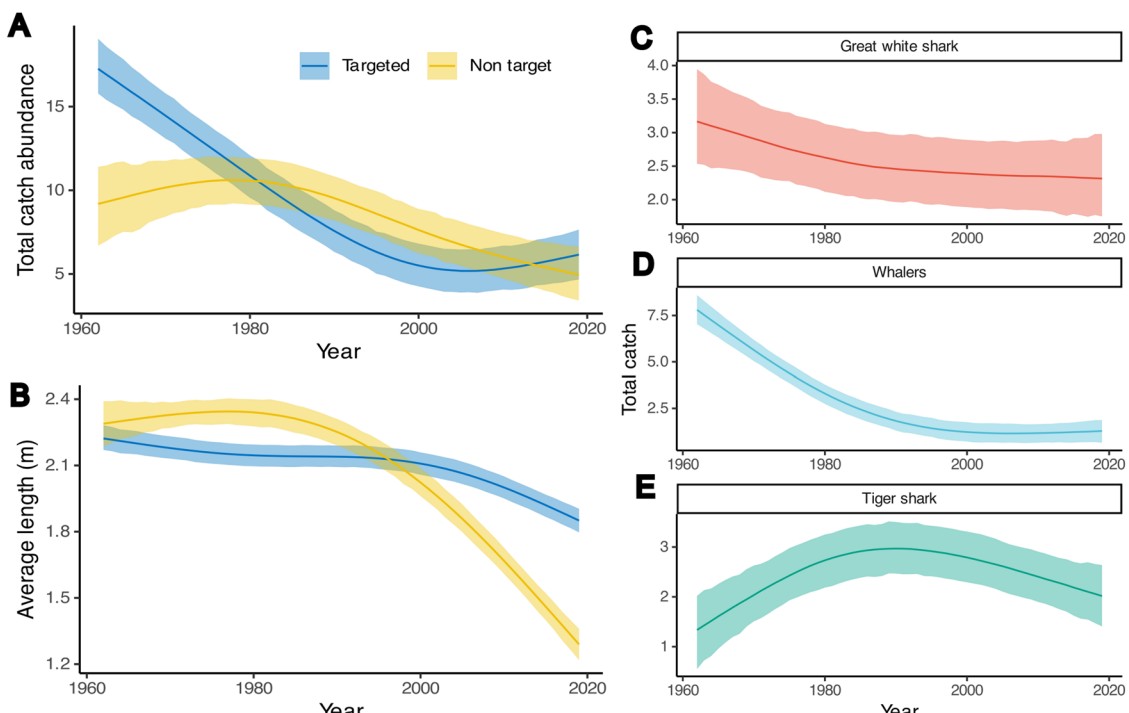

**Fig. 3 | Declines in the abundance of key species and traits.** Bayesian generalised additive mixed models highlights significant changes in (**A**) the total abundance of target and non-target species, (**B**) the average length of target and non-target species, and the abundance of (**C**) great white sharks, (**D**) whaler sharks and (**E**) tiger sharks. Based on *n* = 5336 independent samples.

generalised additive mixed models found that there was a clear increase over time in the functional specialisation of targeted species in the program, suggesting that the targeted species that are caught in the program are more specialist in their functional role in coastal ecosystems (Fig. 4F). For non-target species, Bayesian generalised additive mixed models have shown a decrease in the functional uniqueness of the community, suggesting that the community being caught now may less unique and contain species from across functional trait space (Fig. 4G). The relationships between targeted species and functional uniqueness, and non-targeted species and speciali-sation were not clear over time.

**Long term declines in functional diversity and ecological functioning - what does it mean, and can it recover?**

Consistent declines in targeted apex shark functional richness over time are often associated with the loss of pivotal species and key functional traits from food-webs, and this can greatly diminish ecosystem functioning[3]. However, when this is coupled with an increase over time in the functional diversity of non-target and especially middle trophic order species (e.g., mesopreda-tors), a significant shift in the community has occurred. This indicated the likely presence of an anthropogenically induced trophic cascade[16,18,31]. We report a significant decline in the functional diversity of targeted apex sharks in eastern Australia, as illustrated by reductions in overall trait space and therefore overall ecological functioning. These changes likely reflect the combined impacts of both large scale regional and international harvesting pressure on large and highly mobile fish populations[8,12]. Many of the species that have declined in abundance in eastern Australia are large sharks with large home ranges (10 s to 100 s of kilometres and sometimes larger)[15,32] that are also threatened globally by overharvesting in commercial, recreational and artisanal fisheries[8,16]. Recently, however, there have been reports of some apex shark species increasing in abundance in some parts of the world due to improved regional fisheries management strategies[33]. Locally, it is likely that the QSCP has contributed to changes to shark populations[15,34], however, the number of sharks captured in this program pales in comparison to the annual catches of shark fisheries in different regions around the globe[8,13,18]. Furthermore, the effects of habitat fragmentation and loss, diversifying

coastal disturbances, increased fishing pressure and changes in climate all impact on the structure of coastal fish assemblages at varying spatial scales[2,14,35]. Given the large home ranges of the species caught in the pro-gram, we suggest that the declines in the functional richness of sharks in eastern Australia are likely to be symptomatic of this global impact and only partially explained by the direct removal of apex predators from the QSCP.

We identified consistent reductions in the functional richness of tar-geted apex shark species and increases in the functional richness of non-target species. The finding of reductions in targeted species follow findings elsewhere which highlight the negative influence of the program on coastal sharks and rays[15]. Changes in metrics of functional diversity have been used to predict variation in the delivery of ecological functions, and are often associated with significant impacts to both the structure and functioning of ecosystems[1,4,22]. We quantified a reduction in carrion consumption over time; a crucial ecological function which often tracks the decline in func-tional diversity of apex predators over time. This demonstrates the func-tional consequences of the loss of apex predator sharks and the shifts seen in community composition, and highlights the suitability of using functional diversity to index such changes in ecological functioning in this system. While the functional diversity of targeted apex sharks has declined over time, the functional consequences of the catch are likely cascading to lower order coastal fish[16,18,31]. This may further compound the effects of higher order predator loss on the structure and functioning of key coastal ecosystems[12,15]. A shift in community composition and the removal of top predators from food webs in any environmental realm has dire con-sequences for the functioning of ecosystems and how it responds to disturbance[6,15]. Finally, we found that targeted species which have decreased in functional richness overtime and have experienced a shift in their com-munity composition, are also becoming more functionally specialised overtime, suggesting that coastal ecosystems are at threat of losing sig-nificant components of functional trait space as time continues[14]. The species that declined in abundance in this study exert strong top-down control on food webs and so affect the structure and functioning of eco-systems that are formed by foundation species (e.g., kelp forests, coral reefs and seagrass meadows)[10,15,31]. Such changes in the structure and condition of

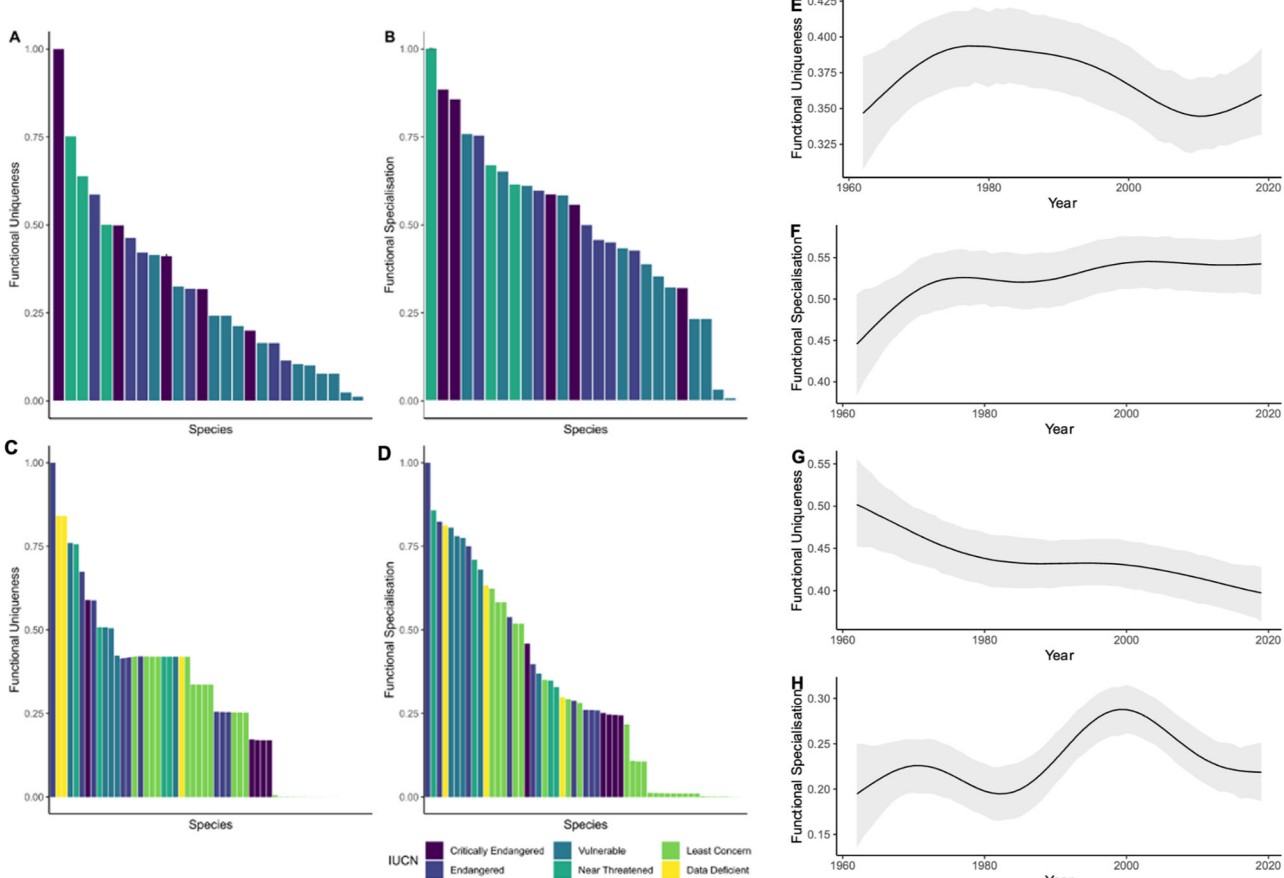

**Fig. 4 | Changes in the specialisation and uniqueness of species overtime.**
Functional uniqueness and specialisation of all target (**A**, **B**) and non-target (**C**, **D**)
species. Typically, the targeted species were more commonly found to be in the
critically endangered to vulnerable categories. All individual functional uniqueness
and specialisation scores are found in Table S2. Additionally, Bayesian GAMMs
found that target species caught in the program increased in functional specialisation
(**F**), while those non-target species caught in the program became more functionally
unique (**G**). We did not find clear patterns for the uniqueness of targeted species (**E**)
or the specialisation (**H**) of non-targets over time.

foundation species have important implications for both ecosystem con-
servation and management including ecosystem support for significant
services such as fisheries, carbon storage and coastal protection[31,36]. These
impacts therefore likely have wide-reaching effects on ecosystem func-
tioning across eastern Australia.

In this study, we use long-term catch records to demonstrate significant
declines in the abundance and changes in the composition of targeted apex
sharks over six decades due to human activities. These changes result in
significant and concerning declines in apex predator functional richness of
coastal ecosystems, and the abundance and functional trait space of most
key targeted functional groups captured in the program. While long-term
datasets quantifying the functions supported by sharks are limited, declines
in the functional richness or diversity of sharks are now being identified in
many ecosystems globally as this type of data is more readily available[14,37].
These long-term declines have wide-reaching ramifications for the condi-
tion of ecosystems and the support of key ecosystem services[3,4]. For example,
declines in large predators have implications for carbon storage and climate
change[38], coastal fisheries[39] and the resilience of ecosystems to multiple
anthropogenic stressors[3]. This effect could be magnified when combined
with synergistic anthropogenic stressors that combine to impact coastal
ecosystems[12,40]. Locally, it is clear from the results that mesh nets are more
harmful, and that this is being increasingly acknowledged by program
managers through reductions in net usage. However, in and of itself, these
actions would not diminish the broader effects of the program on coastal
assemblages and functioning[15]; especially because the effects of a region-
wide trophic cascade are likely significant. More broadly, reducing

overharvesting and indeed recovering the abundance of important func-
tional groups (in this case, targeted coastal apex predators and non-target
species) is crucial. Several key examples exist globally of such changes
resulting in positive outcomes for ecosystems globally, including wolves in
Yellowstone National Park, the protection of otters on the coastline of
California[9] and the protection of sharks in the western Atlantic ocean[33].
Adopting management strategies that restore lost or compressed functional
niches by targeted conservation measures for key species such as sharks is
important[9], as is implementing strategies that can repair or reduce slow,
ongoing, and often passive reductions in functionally important species and
functional groups.

## Methods
### Long-term monitoring from the Queensland Shark Control Program
We analyzed catch data from the Queensland Shark Control Program
(QSCP) in Queensland, Australia, which uses a combination of fixed nets
and baited hooks at up to 80 beaches along 1760 km of coastline. The
primary objective of the program is to reduce risk to swimmers from sharks
by reducing the abundance of large species deemed 'dangerous'. This
includes 19 'dangerous' target species, including *Carcharodon carcharias*,
*Galeocerdo* cuvier and several large *Carcharinidae* species. Over 300 baited
hooks and 27 mesh nets were deployed across Queensland in 2020. At the
peak of the program's effort in 2007, there were approximately 460 baited
hooks and 127 mesh nets set. Data for the program is publicly available and
was accessed from the Queensland Department of Agriculture and Fisheries

(https://www.data.qld.gov.au/dataset/shark-control-program-shark-catch-statistics). The program uses 186 m long and 6 m drop nets with 50 cm mesh size that are typically deployed parallel to the beach in 7–12 m water depth and 500–1000 m from the beach. Drumlines consist of a baited 14/0 J hook, historically baited with shark flesh, but sea mullet *Mugil cephalis* is the most common bait today. These are also typically deployed 500–1000 m from shore. Mesh nets and baited drumlines are checked by contractors approximately 20 days per month[41]. While mesh nets were the predominate method used in the early days of the program, extensive bycatch led to a gradual shift in effort to baited drumlines over time (see statistical analysis section).

### Functional diversity analysis

Functional richness[42–45] was indexed as variation in traits of targeted shark and non-targeted species. Traits for each species captured between 1962 and 2019 in the QSCP were extracted from either FishBase (all morphological traits, feeding group and habitat preference) or were based on information in the literature (e.g., movement scale)[46]. See Table S1 for a full list of all traits, their hypothesised links to functioning and their definition and Supplementary Data 1 for the trait values and categories for each species. We used two sets of traits (ecological and morphological) to characterise functional richness with each set of traits indexing the functional role (i.e., habitat preference, feeding group, movement scale) and general morphology (i.e., maximum total length, eye diameter, head length) of species[47]. Each of these traits are important in determining the role that a species has in the broader coastal ecosystem, the scale of that role and the morphological differences between species. Feeding groups included invertebrate, small-bodied fish feeding, large-bodied fish feeding and megafauna feeding. Habitat preference included reef associated, coastal pelagic, oceanic pelagic, coastal benthic and oceanic benthic. Movement scale was broken up into three categories, 0–100 km, 100–500 km and greater than 500 km. Teeth morphology was broken up into three categories, triangular, angular and grinding plates. All morphological metrics were continuous measures. The combination of both categorical and continuous functional trait measures allowed us to specifically identify the functional niche that the different species are operating in, while using the morphology traits to tease out the differences between functionally similar species. For example, by relying only on ecological traits to partition the species within our overall community, we would miss some of the nuances between individual species that may feed in subtly different ways[11,48].

Functional richness was calculated with the *fundiv* and *FD* packages in R[21,43,44]. Functional richness ($FR_{ci}$)[49] quantifies the area of functional trait space in an ecosystem[44]. Here, $SF_{ci}$ is the niche space filled by the species within the community and $R_c$ is the absolute range as set out by the species.

$$FR_{ci} = \frac{SF_{ci}}{R_c} \qquad (1)$$

Previous studies investigating patterns in the QSCP identified issues with species identification in data prior to 1997, where many species such as whalers were seldom identified to species level[15]. To account for this, we classified all whaler and hammerhead catches (regardless of the level of identification) into broader taxonomic groups and assigned them an average trait value for continuous traits and the most common category for categorical traits for all whaler and hammerhead species respectively combined.

### Statistical analysis and reproducibility

Data was reorganised to represent the abundance of catch of all species at each beach, for each gear type for each year of the program (i.e., from 1962 to 2019). Sampling locations within each region are provided in the QSCP database and refer to each different locality where shark nets and drumlines are deployed. We combined our catch across the whole year to account for the effect that season has on the types of species and the abundance in which they were caught. We then split the community into target and non-target species, to identify variation in functional diversity between the two groups over time.

We used Bayesian generalised additive mixed models (GAMMs) in the package *brms* and *rstan* to assess the relationships between functional richness and time[50]. Time (year) and gear type (nets and baited drumlines) were treated as fixed factors in our analysis, to account for the differences in the catchability between the different gear types over time. To account for changes in the number of nets and baited drumlines used in the program per year, we included the total number of mesh nets and baited drumlines at each beach for each year as a fixed factor in the analysis, therefore accounting for variation in effort over the period of the program. We included region and location (i.e., the name of an individual beach) nested within region to account for site-specific effects. All models were tested to ensure they met assumptions of normality using a *qqplot* and by checking the variation associated with the random effect of location. Best fit models were those with parameters selected that were given broad priors (normal priors with a mean of 0 and a standard deviation of 4) and were calculated using four Markov chains with 4000 iterations (1000 iteration warmup). All r hat values for Bayesian GAMMs were between 1 and 1.01 indicating model convergence. We assessed changes in ecological functioning over time by determining the number of captures per drumline per year. While this may be a crude estimate of functioning because it ignores carrion consumption that is occurring at lower trophic levels and by non-fish species, it is still indicative of the levels of carrion consumption in the system. As stated by the QSCP, drumlines are checked on average 20 days per month, which allowed us to determine a rate of scavenging. This was done by taking the total number of captures on drumlines at a single beach and dividing that by number of days a drumline is checked and then by the overall effort for that beach (e.g., number of drumlines set per year). We used proportion of captures as information on the removal of baits alone is not made public. Further, this is likely quantified differently between individual contractors, making bait removal measures unreliable for this analysis. We used Bayesian GAMMs to determine how ecological functioning changed over time and relative to changes in functional richness using the same model structure as other Bayesian GAMMs.

We used a multivariate generalised additive mixed model (GAMM) in the *mvabund* package in R to test for the effects of year on community composition. This was completed using the *manyany* function and calculated models with and without the variable year to determine the effect of time[51]. Community composition was visualised using a PCoA plot with all sites within a year being averaged for each of nets and drumlines. We calculated the species specific metrics functional uniqueness (FUn) and functional specialisation (FSp) using the *fuse* function in the *mfd* package[52]. To calculate the FUn and FSp metrics for individual communities over time, we used the individual scores and weighted these by the presence and abundance of each species, within each community. We used Bayesian GAMMs (with the above-described model structure) to assess changes in the abundance of all species caught, changes in length and the abundance of great white sharks, tiger sharks and whalers, and the functional uniqueness and specialisation of targeted and non-target species through time.

### Reporting summary

Further information on research design is available in the Nature Portfolio Reporting Summary linked to this article.

### Data availability

The shark control data and associated effort data is courtesy of the Queensland Department of Agriculture and Fisheries, Australia. The data is publicly available upon request from the Queensland Department of Agriculture and Fisheries. All ecological and morphological trait data is available in Supplementary Data 1. All code used in the analysis is available at the following GitHub repository, https://github.com/chrishendersonUSC/multi_decadal_collapse_QSCP.

**Article**

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

## Acknowledgements
The authors would like to acknowledge George Roff for his helpful comments on statistical analysis and the manuscript.

## Author contributions
C.H., B.G., and A.O. conceived the idea. C.H., B.G., L.G.G., G.R., M.T., J.M. led the analysis and C.H. led the writing of the manuscript. C.H., B.G., M.T., L.G.G., J.M., T.S., H.B., and A.O. were involved in the drafting of the manuscript.

## Competing interests
The authors declare no competing interests.
