## [Peer review file · Communications Biology]

Reviewers' comments:

Reviewer #1 (Remarks to the Author):

This manuscript aims to understand long-term ecological effects of shark declines along the eastern Australian coast due to drum lines and mesh nets. A decline in functional diversity is identified with regards to large species targeted by nets and drum lines. This is a relevant study due to both the worldwide declines of sharks; as well as recent findings that shark functional diversity is acutely vulnerable. However, I think the study could also do with some additional tests to make its evidence stronger, and to provide some additional angles. I also think some of the writing could be improved by providing clearer criteria for the study's approach (see major comments below), as it currently seems a bit vague and might be hard to follow for readers who aren't as caught up in the literature.

Main comments

- The title could be modified to refer to long term functional diversity declines in Eastern Australian coastal waters; seeing as the study has a clear focus on this region.
- Throughout the manuscript, I think it would benefit the study to be more direct with exactly how sharks are threatened (e.g., L32-33 in the introduction). A study by Dulvy et al. (2021) will be most useful as it shows that up to a third of shark species are threatened with extinction, with overfishing being the overwhelming primary cause.
- It's not fully clear what criteria is being used to define "large/apex" or "middle trophic order" (L171) sharks.
- Please briefly justify the use of traits.
- I think it would be useful to show the functional trait space of the sharks used in this study in 1962 and 2019 to visualise changes in functional richness for each group of sharks. An analysis of how many axes best represent the data and which traits correlate most with each axis should also be provided here, as this is standard practise when constructing a trait space. The mFD R package (Magneville et al. 2022) provides a clear and helpful tutorial for all the necessary steps that should be of use to the authors here.
- It's not clear to me where the results of the "probability of an effect" (L97/L104) are coming from. Which statistical test, if any, produced these results? A statistical test to quantify the "significant changes" in abundance and average length (L153/Figure 3) would also be appreciated to support the findings even more.
- It's not stated why the key traits and species illustrated in Figure 3 are key. Please provide the criteria.

Minor comments

Consider calculating the FUSE metric, as well as changes in functional uniqueness and specialisation over time, as this can reveal if the assemblage has become more vulnerable to species loss or highly specialised.

Figure 2: What % of the total inertia (or variance) is provided by PCOA 1 and 2?

L30: Please provide some additional context as to why sharks are such important components of marine food webs.

L44: Unclear what the "rate" of an ecological function is in context of the study.

L123-124: Redundant to say "scavenge opportunistically when the opportunity arises".

L155: "bu" I assume is a typo for "by".

L163-164: Could you please give an example of the "important implications for both ecosystem conservation and management"?

L167: Please change "is" to "are"

L185: Please provide references for the "expansive home ranges" of caught species. One useful

example would be Bonfil et al. 2005 documenting transoceanic migrations in the great white shark.

L206: Please change “this actions” to “these actions”.

L256-258: I think moving these three sentences to the end of the prior paragraph will benefit the flow of the narrative.

L264: Do you have a reference for the loss of individual species nuances that comes from relying on only ecological traits?

Table S2: Could you please include the scientific names of all species?

Supplementary material: Could you please include a reference list for all the useful references included in Table S1?

Reviewer #2 (Remarks to the Author):

This paper utilizes a long-term dataset to investigate trends in coastal shark populations (abundance and a suite of morphological/ecological traits) in Australia. The data set and many of the current analyses are interesting, but as presented, the manuscript leaves much to be desired.

First, despite the title and much text, very little is presented in terms of trait-based diversity in the results. Length is considered briefly (Fig 3), but where are the rest of the trait results beyond their explanation in table S2 in the supplement? The trait-based component was the particularly novel aspect of this study. Much more is focused on abundance.

Second, much of the writing needs to be overhauled. The Results and Discussion are highly redundant. Other comments are provided in the specific comments below.

Finally, there are 2 critical analytical issues to be addressed. First, the potential for changes in effort to impact results (abundance and species/trait diversity). While the authors include effort as a fixed effect in their modeling, the results of that analysis are never provided. Much more needs to be included regarding the impacts of effort on all of the results. I would recommend modifying the analytical approach to correct for effort on the front end (ie, standardized by hooks or net sets) rather than include that as a factor in the modeling. Next, I'm skeptical of the link between capture rate and scavenging function. To start, the authors need to be much more clear about how they are defining this ecological functioning metric. Is it baits removed, or sharks on hooks? These are critically different because catchability of smaller species (which may be more likely to scavenge) is likely different and could have changed over time.

Specific comments

Title: I recommend revising the title somewhat. As written, it is overly broad and generalizes (across spatial scales) beyond the scale supported by the data from one survey.

Introduction:

L30: Revise this sentence, cut fluff words, and provide something beyond “public appeal and interest”, which is not relevant to their diversity or role within ecosystems.

L36: what are support functions?

L36-37: Evolving to become evolutionarily distinct is circular. Revise

L49-52: Revise this sentence for clarity. First, ‘ecology of an organism’ is overly vague; second, clarify what you mean regarding size and direct vs indirect effects.

L55-56: Regarding management applications, either expand on this or remove mention of

management.

L56-59: Regarding the term multifunctionality, please revise for clarity. This bit is repetitive and jargon heavy.

L78-80: Odd placement for this sentence, which follows all of the previous text in this paragraph where the findings are revealed. I suggest revising this final paragraph and either removing the hypotheses or moving the first sentence (which summarizes the results) to the end of the paragraph.

Methods:

L245-246: Revise to remove duplicate naming conventions.

L250: Confirm, 3 or 4 groups?

L272-8: Need to provide additional justification that this potential mis-ID is not impacting results. This issue is re-visited in the Discussion (L140) but no additional justification is provided there either.

L281: What does data reconstruction mean?

L304-306: How does number of captures assess changes in ecological functioning? Revise.

L306-312: All of this needs to be clarified.

Results & Discussion:

L84-95: First paragraph is not Results. Move into Methods section or Introduction.

L114: What evidence can you point to that 'functional consequences of the catch are likely cascading to lower order fish'?

L 122 & Fig 1B: Need to clarify, is this proportion of baits taken, or number of hooks with a shark? Because bait could be removed but not shark is captured, which would be a critical difference.

Fig 1C: This figure is not at all clear, nor does the associated text provide a clear interpretation. Revise this set of results.

L97-144: This seems to all be redundant. Revise this section to be more concise and provide clear interpretation of your results.

L169: where is resilience measured? Remove this word throughout- no measures of resilience were made.

L175-182: This is all well and good, but one could point to other studies demonstrating a rebound in sharks over the past 10+ years. The dichotomy between the results here and from elsewhere merits more discussion.

Response to reviewers

Thank you to the editor and two anonymous reviewers for their comments on the manuscript. These have certainly resulted in a much better manuscript. Please find responses to the comments below, with all changes to the manuscript highlighted in red.

Comment number	Line number	Comment	Response
Reviewer 1			
1		This manuscript aims to understand long-term ecological effects of shark declines along the eastern Australian coast due to drum lines and mesh nets. A decline in functional diversity is identified with regards to large species targeted by nets and drum lines. This is a relevant study due to both the worldwide declines of sharks; as well as recent findings that shark functional diversity is acutely vulnerable. However, I think the study could also do with some additional tests to make its evidence stronger, and to provide some additional angles. I also think some of the writing could be improved by providing clearer criteria for the study's approach (see major comments below), as it currently seems a bit vague and might be hard to follow for readers who aren't as caught up in the literature.	Thank you for your helpful comments throughout the review. We have addressed these below and appreciate the improvement this has led to for the manuscript.
2	Title	The title could be modified to refer to long term functional diversity declines in Eastern Australian coastal waters; seeing as the study has a clear focus on this region.	We have modified the title to reflect this suggestion. "Long term declines in the functional diversity of sharks in the coastal oceans of eastern Australia"
3	Ln 32	Throughout the manuscript, I think it would benefit the study to be more direct with exactly how sharks are threatened (e.g., L32-33 in the introduction). A study by Dulvy et al. (2021) will be most useful as it shows that up to a third of shark species are threatened with extinction, with overfishing being the overwhelming primary cause.	We have modified this sentence based on the reviewer suggestion. "However, they are experiencing significant threats and declining due to human activities (e.g. overharvesting, climate change, habitat loss) ^{8,12-16} , with overfishing alone driving one third of shark and ray species to extinction ¹⁷ ".

4	Ln 70	It's not fully clear what criteria is being used to define "large/apex" or "middle trophic order" (L171) sharks.	We have updated this in Ln 71 to be more consistent in our language throughout. "Whilst the primary intent of the program is to remove large shark species (hereafter referred to as targeted apex shark species) thought to pose risks to swimmers, approximately 75% of species caught in the program are incidental catches." Therefore, we are referring to those species that are the targets of the program as they have been deemed to be a risk to swimmers.
5	Ln 68 - 94	Please briefly justify the use of traits.	We have modified the final paragraph of the introduction in line with this comment and one from reviewer 2. This provides a better justification for the use of traits in this study. "We use data from the fisheries-independent Queensland Shark Control Program (QSCP), which has deployed a combination of mesh nets and baited hooks along 1,760 km of coastline continuously since 1962¹⁵. Whilst the primary intent of the program is to remove large shark species (hereafter referred to as targeted apex shark species) thought to pose risks to swimmers, approximately 75% of species caught in the program are incidental catches. Because the field implementation of the program is standardised, and it incidentally captures many non-target fish species, it effectively represents a high density, high frequency, and long-term sampling of higher-order coastal fishes. It offers a long-term record of the continual capture of sharks, rays and other teleosts along the Queensland coastline with a known effort¹⁵. We used this publicly available long-term catch data to demonstrate fundamental changes in the functional diversity, ecological functioning and species composition of targeted and non-targeted

			coastal species over six decades. Traditional measures which quantify change in the abundance and diversity of species such as species richness are rarely effective for quantifying the loss of functional roles in ecosystems^{20,26,27}. Trait-based approaches are considered the most appropriate when effective data describing ecological functioning is not available. Therefore, traits are widely and increasingly used to assess ecosystem functioning change in ecological and evolutionary studies^{20,21}. We characterise the functional traits of targeted apex sharks and non-targeted coastal fish species using two different approaches to calculate functional diversity. These two approaches included two separate types of trait values; one quantifying change in ecological traits (e.g. feeding group, habitat preference and movement scale) and the other quantifying change in morphological traits (e.g. maximum total length, eye diameter, teeth morphology)^{28,29}. We used these two approaches because we wanted to understand the effects of the shark control program from multiple functional perspectives. We hypothesise that widespread reductions in the numbers of targeted apex shark species will result in significant declines of functional diversity and ecological functioning over time^{8,15}.”
6	Results	I think it would be useful to show the functional trait space of the sharks used in this study in 1962 and 2019 to visualise changes in functional richness for each group of sharks. An analysis of how many axes best represent the data and which traits correlate most with each axis should also be provided here, as this is standard practise when constructing a trait space. The mFD R package (Magneville et al. 2022) provides a clear and helpful tutorial for all the necessary steps that should be of use to the authors here.	The mFD package does indeed show this nicely, but in this instance, such an illustration does not adequately represent the data because it only shows two small snapshots of the change in functional trait space would not be useful to show the effect of the program as we have communities from up to 30 beaches across 7 regions over a 60 year period, so these single snapshots are not the best representation of what is happening.

7	Results, Ln 100 and throughout	It's not clear to me where the results of the "probability of an effect" (L97/L104) are coming from. Which statistical test, if any, produced these results? A statistical test to quantify the "significant changes" in abundance and average length (L153/Figure 3) would also be appreciated to support the findings even more.	Throughout this study, we used Bayesian generalised additive mixed models. With Bayesian statistics, rather than the output producing an r value, the output is a probability of an effect. We have updated the sentences that introduce this to highlight this point. "Bayesian generalised additive mixed models show that there was a 99.88% and 100% probability of an effect to suggest that the functional richness of targeted sharks based on ecological and morphological traits declined, respectively, over the last six decades (Fig. 1A)."
8	Fig 3	It's not stated why the key traits and species illustrated in Figure 3 are key. Please provide the criteria.	We have updated the text to highlight why we used these community traits and species. The figure caption is Figure 3. Shifts in ecosystem attributes and declines in the abundance of target species. Bayesian generalised additive mixed models highlights significant changes in ecosystem attributes (A) the total abundance of target and non-target species, (B) the average length of target and non-target species, and the abundance of (C) great white sharks, (D) whaler sharks and (E) tiger sharks
9	Ln 165 – 179, 226	Consider calculating the FUSE metric, as well as changes in functional uniqueness and specialisation over time, as this can reveal if the assemblage has become more vulnerable to species loss or highly specialised.	We have added the fuse metrics to the paper, which includes additional text in the results, discussion, an additional figure in the main document and an additional table in the supplementary materials. We have added the following text to the results. "Shifts in the functional uniqueness and specialisation of species Functional uniqueness and specialisation metrics are suitable for assessing the levels of redundancy in the community and whether species in the community are more generalist or specialist in nature. We found that typically, targeted species that had higher levels of uniqueness and specialisation (Fig. 4A-B, Table S3) and were typically more

			endangered, while non-targeted species were from a variety of threatened categories (Fig. 4C-D, Table S3). Bayesian GAMMs found that there was a clear increase over time in the functional specialisation of targeted species in the program, suggesting that the targeted species that are caught in the program are more specialist in their functional role in coastal ecosystems (Fig. 4F). For non-target species, Bayesian GAMMs have shown a decrease in the functional uniqueness of the community, suggesting that the community being caught now may be less unique and contain species from across functional trait space (Fig. 4G). The relationships between targeted species and functional uniqueness, and non-targeted species and specialisation were not clear over time.” We have also added the following sentence to the discussion paragraphs “Finally, we found that targeted species which have decreased in functional richness overtime and have experienced a shift in their community composition, are also becoming more functionally specialised overtime, suggesting that coastal ecosystems are at threat of losing significant components of functional trait space as time continues¹⁴.”
10	Fig 2	Figure 2: What % of the total inertia (or variance) is provided by PCOA 1 and 2?	The method that we have used to calculate PCOA1 and 2 in our analysis is based on the function cmdscale in R. To determine variation explained by each axis, we calculated a correlation between PCOA1 and PCOA2 against the matrix of our community. This resulted in each axis explaining between 24% and 27.5%. This has been updated in the Figure 2 caption.

			“Fig. 2. Significant shifts in community composition over time. A principle coordinates analysis plot highlighting the changes in the composition of targeted species in the Queensland Shark Control Program from 1962 to 2019. The change in composition is significant for both drumlines and nets, and targeted and non-targeted species with a clear shift in each community. Points in each plot represent centroids for each year, with the colour of that point going from dark blue (1962) to light blue (2019). PCOA 1 explained approximately 24.3% of the variation and PCOA 2 27.5%.”
11	Ln 30	L30: Please provide some additional context as to why sharks are such important components of marine food webs.	We have modified this sentence to provide further context. “Sharks are functionally important components of coastal and oceanic food webs as they exert top-down pressure on food webs at large spatial scales through the direct and indirect effect of predation. They are also well researched, meaning that their abundance and diversity if well understood in many settings”
12	Ln 47	L44: Unclear what the “rate” of an ecological function is in context of the study.	We have clarified this in the below sentence “The rate (e.g. the frequency or quantity) and distribution (e.g. the spatial scale) of ecological functions across landscapes are intrinsically linked to biodiversity because a greater variety of species performing a particular function will usually increase both the rate and stability of that function.”
13	Ln 133	L123-124: Redundant to say “scavenge opportunistically when the opportunity arises”.	This has been updated to remove the redundant wording.
14	Ln 156	L155: “bu” I assume is a typo for “by”.	Updated
15	Ln 223	L163-164: Could you please give an example of the “important implications for both ecosystem conservation and management”?	We have updated this sentence to include examples “Such changes in the structure and condition of foundation species have important implications for both ecosystem conservation and management including ecosystem support for services such as fisheries, carbon storage and coastal protection.”

16		L167: Please change “is” to “are”	Updated
17	Ln 193	L185: Please provide references for the “expansive home ranges” of caught species. One useful example would be Bonfil et al. 2005 documenting transoceanic migrations in the great white shark.	We have added two references here to support this statement. Thank you for the suggestion.
18		L206: Please change “this actions” to “these actions”.	Updated
19	Ln 305	L256-258: I think moving these three sentences to the end of the prior paragraph will benefit the flow of the narrative.	This has been updated to reflect this suggestion
20	Ln 312	L264: Do you have a reference for the loss of individual species nuances that comes from relying on only ecological traits?	We have added two references to this statement to further support the importance of traits. Pimiento, C. et al. Functional diversity of sharks and rays is highly vulnerable and supported by unique species and locations worldwide. Nature Communications 14 , 7691 (2023). https://doi.org:10.1038/s41467-023-43212-3 Cadotte, M. W. Functional traits explain ecosystem function through opposing mechanisms. Ecology letters 20 , 989-996 (2017).
21	Table S2	Table S2: Could you please include the scientific names of all species?	We have included these throughout.
22	Supps	Supplementary material: Could you please include a reference list for all the useful references included in Table S1?	This has been included in the supplementary materials.
Reviewer 2			
23		This paper utilizes a long-term dataset to investigate trends in coastal shark populations (abundance and a suite of morphological/ecological traits) in Australia. The data set and many of the current analyses are interesting, but as presented, the manuscript leaves much to be desired.	We thank the reviewer for their comments. Our edits according to their suggestions have significantly improved the manuscript.
24	Fig 4, Ln 165 - 179	First, despite the title and much text, very little is presented in terms of trait-based diversity in the	Thank you for your comment here. We completely agree with this assessment of the changes in traits over time.

		results. Length is considered briefly (Fig 3), but where are the rest of the trait results beyond their explanation in table S2 in the supplement? The trait-based component was the particularly novel aspect of this study. Much more is focused on abundance.	Unfortunately, the only trait the is measured on individual captures throughout the program is length of individual caught. Because we have this information, we were able to show how this trait changes over time, but for the remaining traits, we rely on the measure of functional richness to show trait change through time, but the value of each trait for each individual species does not change from year to year. Regarding trait based diversity analyses overall, we have added additional analysis suggested by reviewer 1 which highlights the changes overtime in the functional uniqueness and specialisation of species caught in the program.
25	Results and discussion	Second, much of the writing needs to be overhauled. The Results and Discussion are highly redundant. Other comments are provided in the specific comments below.	We have made significant changes throughout the results and discussion sections of the manuscript with the aim of removing redundant content and streamlining the section. This is highlighted through comments below and in the structure of the text.
26	Ln 117 - 147	Finally, there are 2 critical analytical issues to be addressed. First, the potential for changes in effort to impact results (abundance and species/trait diversity). While the authors include effort as a fixed effect in their modelling, the results of that analysis are never provided. Much more needs to be included regarding the impacts of effort on all of the results. I would recommend modifying the analytical approach to correct for effort on the front end (ie, standardized by hooks or net sets) rather than include that as a factor in the modelling. Next, I'm skeptical of the link between capture rate and scavenging function. To start, the authors need to be much clearer about how they are defining this ecological functioning metric. Is it baits removed, or sharks on hooks? These are critically different because catchability of smaller species (which may	We have made significant changes to the text associated with both effort and the measure of carrion consumption to provide further details on these measures and effects. This includes the addition of two supplementary figures showing how removing the grouping of species effect on predators alters the effects of functional richness over time and how effort changes over time for each functional richness measure. Regarding the reviewer comment about changing our metric to be standardised for effort rather than including it as a fixed factor, we disagree that this is a more appropriate way of assessing effort impacts. We have included the effect of effort in the model as a fixed factor to account for the variables in the model rather than using CPUE because the measure of functional richness is not weighted by abundance and therefore having these values be CPUE rather than total abundance will not have an effect.

		be more likely to scavenge) is likely different and could have changed over time.	“Prior to 1997, poorer identification classifications resulted in a higher number of individuals identified to genus level (e.g. ‘whaler’). To account for this, we averaged the functional traits of all genus or groups within which an individual was identified because we wanted to limit the effects of new species being identified in the community after this time (e.g. those that were previously mis-identified), even though the species was likely they have been caught previously . Therefore, we analysed the same effects on all targeted apex sharks without them being grouped to show that these patterns hold true (See Fig. S2). Crucially, the catch effort at and between individual beaches has not been consistent over the life of the program. Effort was included in all models assessing change in functional richness, with Bayesian generalized additive mixed models on functional richness there was always a greater than 90% chance of an effect of effort on functional richness. The effect of effort on the functional richness of targeted and non-targeted species was typically non-linear using both ecological and morphological traits, with functional richness typically reaching its maximum at low levels of effort and then maintaining that level as effort increased (See Fig. S3). “We assessed changes in ecological functioning over time by determining the number of captures per drumline per year. While this may be a crude estimate of functioning, it is still indicative of the levels of carrion consumption in the system, but it does, however, ignore carrion consumption that is occurring at lower trophic levels and by non-fish species. As stated by the QSCP, drumlines are checked on average 20 days per month, which allowed us to determine a rate of scavenging. This was done by taking the total number of captures on drumlines at a single beach and dividing that by number of days a drumline is checked and
--	--	--	---

			then by the overall effort for that beach (e.g. number of drumlines set per year). We used proportion of captures as information on the removal of baits alone is not made public. Further, this is likely quantified differently between individual contractors, making bait removal measures unreliable for this analysis. "
27	Title	Title: I recommend revising the title somewhat. As written, it is overly broad and generalizes (across spatial scales) beyond the scale supported by the data from one survey.	We have updated the title to reflect suggestions from Reviewer 1. "Long term declines in the functional diversity of sharks in the coastal oceans of eastern Australia "
28	Ln 33.	L30: Revise this sentence, cut fluff words, and provide something beyond "public appeal and interest", which is not relevant to their diversity or role within ecosystems.	We have modified this sentence "However, they are experiencing significant threats and declining due to human activities (e.g. overharvesting, climate change, habitat loss), with overfishing alone driving one third of shark and ray species to extinction. "
29	Ln 38	L36: what are support functions?	We have updated this sentence. "Maintaining a diversity of traits within an ecosystem is crucial to ensure the continued provision of key ecological functions across the food web "
30	Ln 40	L36-37: Evolving to become evolutionarily distinct is circular. Revise	We have revised to remove the circular statement.
31	Ln 53	L49-52: Revise this sentence for clarity. First, 'ecology of an organism' is overly vague; second, clarify what you mean regarding size and direct vs indirect effects.	We have clarified this sentence by highlight what aspects of the ecology of a species' traits can be used to quantify. Here we use examples such as differentiating the role of a species, where they feed or the size of that individual. "Traits are useful as they quantify the ecology of an organism by differentiating the role a species plays in an ecosystem, such as how and where they feed or the size of an individual species"

32	Ln 57	L55-56: Regarding management applications, either expand on this or remove mention of management.	We have expanded this sentence to reflect this comment “Furthermore, understanding the functional traits of coastal predators such as sharks is crucial for coastal management as shark play a key role in the coastal food web and their loss is likely to have substantial consequences for the structure and functioning of that ecosystem.”
33	Ln 61	L56-59: Regarding the term multifunctionality, please revise for clarity. This bit is repetitive and jargon heavy.	We have modified this sentence to reduce jargon and highlight how functional diversity as a metric is linked theoretically to the provision of multiple ecological functions. “Changes in functional diversity, a metric that quantifies variation in the set of traits possessed by all species in a community, is often linked to changes in the provision of multiple ecological functions linked to resilience in an ecosystem”
34	Ln 68 - 94	L78-80: Odd placement for this sentence, which follows all of the previous text in this paragraph where the findings are revealed. I suggest revising this final paragraph and either removing the hypotheses or moving the first sentence (which summarizes the results) to the end of the paragraph.	Thank you for this suggestion. We agree and have moved this to the end of the paragraph prior to the results.
35		L245-246: Revise to remove duplicate naming conventions.	We have removed the common names here
36	Ln 301	L250: Confirm, 3 or 4 groups?	This has been rewritten to reflect this comment “Feeding groups included invertebrate, small-bodied fish feeding, large-bodied fish feeding and megafauna feeding”
37	Ln 118	L272-8: Need to provide additional justification that this potential mis-ID is not impacting results. This issue is re-visited in the Discussion (L140) but no additional justification is provided there either.	We have provided more detail in the results and discussion section on this and have provided additional analysis on the community without whalers and hammerheads being combined. This figure is now Supplementary Figure 2.

			“To account for this, we averaged the functional traits of all genus or groups within which an individual was identified because we wanted to limit the effects of new species being identified in the community after this time, even though it was likely they have previously been caught. We did, however, analyse the same effects on all targeted apex sharks without them being grouped to show that these patterns do hold true (See Fig. S2).”
38	Ln 269	L281: What does data reconstruction mean?	We have moved the word reconstructed to remove any confusion. The raw data has simply been reorganised to suit the analyses done.
39	Ln 352	L304-306: How does number of captures assess changes in ecological functioning? Revise.	We have added more text here and to the results/discussion to further explain the use of this metric. “We assessed changes in ecological functioning over time by determining the number of captures per drumline per year. While this may be a crude estimate of functioning because it ignores carrion consumption that is occurring at lower trophic levels and by non-fish species, it is still indicative of the levels of carrion consumption in the system. As stated by the QSCP, drumlines are checked on average 20 days per month, which allowed us to determine a rate of scavenging. This was done by taking the total number of captures on drumlines at a single beach and dividing that by number of days a drumline is checked and then by the overall effort for that beach (e.g. number of drumlines set per year). We used proportion of captures as information on the removal of baits alone is not made public and is likely to be different between individual contractors.”
40	Ln 352	L306-312: All of this needs to be clarified.	Please see the response to comment 39.
41		L84-95: First paragraph is not Results. Move into Methods section or Introduction.	We have removed this first paragraph due to comments from both reviewers.
42	Ln 222	L114: What evidence can you point to that ‘functional consequences of the catch are likely cascading to lower order fish’?	We have added three references to this statement to provide further evidence.

			Baum, J. K. & Worm, B. Cascading top-down effects of changing oceanic predator abundances. J. Anim. Ecol. 78, 699-714 (2009). Myers, R. A., Baum, J. K., Shepherd, T. D., Powers, S. P. & Peterson, C. H. Cascading effects of the loss of apex predatory sharks from a coastal ocean. Science 315, 1846-1850 (2007). Ward, P. & Myers, R. A. Shifts in open-ocean fish communities coinciding with the commencement of commercial fishing Ecology 86, 835-847 (2005). https://doi.org:10.1890/03-0746
43	Ln 136	L 122 & Fig 1B: Need to clarify, is this proportion of baits taken, or number of hooks with a shark? Because bait could be removed but not shark is captured, which would be a critical difference.	We have added text to clarify what this measure is and the limitations that may be present with this method. “We used proportion of captures in this case, as information on the removal of baits alone is not included in the public database and would be different between individual contractors. While this may be a crude metric of actual carrion consumption, as the full information on bait replacement is not kept, this is the most accurate, but likely still an underestimate, measure of function that can be identified in the current database.”
44	Ln 144	Fig 1C: This figure is not at all clear, nor does the associated text provide a clear interpretation. Revise this set of results.	We have added more text to provide more detail on this aspect of the figure “Here, we show that ecological functioning and functional richness were highest at the beginning of the program. Functioning, here carrion consumption, remains high when functional richness is high, but this slightly decreased as time continued (Fig. 1C).”

45	Results and discussion	L97-144: This seems to all be redundant. Revise this section to be more concise and provide clear interpretation of your results.	We have made significant changes throughout the results and discussion sections to reduce redundant content and wording.
46		L169: where is resilience measured? Remove this word throughout- no measures of resilience were made.	We have removed the word resilience from the results and discussion section of the manuscript to avoid any confusion with what we have measured.
47	Ln 193	L175-182: This is all well and good, but one could point to other studies demonstrating a rebound in sharks over the past 10+ years. The dichotomy between the results here and from elsewhere merits more discussion.	We have added additional information to further highlight that some species are seeing a reverse in these trends in some aspects of the world. “Many of the species that have declined in abundance in eastern Australia are large sharks with expansive home ranges^{15,34} that are also threatened globally by overharvesting in commercial, recreational and artisanal fisheries (Fig. 3C-E)^{8,16}. Recently, however, there have been reports of some apex shark species increasing in abundance in some parts of the world due to improved regional fisheries management strategies³⁵. Locally, it is likely that the QSCP has contributed to changes in shark populations^{15,36}, however, the number of sharks captured in this program pales in comparison to the annual catches of shark fisheries in different regions around the globe^{8,13,18}.”

REVIEWERS' COMMENTS:

Reviewer #1 (Remarks to the Author):

Many thanks for the opportunity to re-review this manuscript. It is my view that the authors have taken great care to address my comments from the last round. I particularly appreciate the additional discussion of functional uniqueness and specialisation, with a very nice additional figure, which has strengthened the manuscript and its conclusions in my opinion. As such, I am satisfied with the revised version of the manuscript and believe it can be published, barring a few minor revisions suggested below. I do not believe I will need to see the manuscript again before publication.

Congratulations to the authors on their interesting work. I look forward to seeing this paper published in the near future.

Minor comments:

L14: Maybe refer to "large sharks" as being apex and targeted here in the abstract for sake of consistency with the rest of the manuscript.

L33: I believe "if" should be "is".

L59: I assume it's meant to be "sharks play a key role..."

L121: "even though species was likely they have been caught previously" is a bit strangely written; I'm assuming this is meant to be something like "species was likely to have been previously caught".

L168: I'd like to see brief definitions given to functional uniqueness and functional specialisation, as done for functional richness earlier, so that their full context is clear to a reader. This may grant opportunity for extra discussion about the susceptibility of the assemblage to further losses too.

L374: Please state which function was used in the mFD package to calculate functional uniqueness and specialisation. I'm presuming it's the "fuse" function, but a reader may not necessarily have experience with this package. Thanks.

Response to reviewers

Thank you to the editor and two anonymous reviewers for their comments on the manuscript. These have certainly resulted in a much better manuscript. Please find responses to the comments below, with all changes to the manuscript highlighted in red.

Comment number	Line number	Comment	Response
Reviewer 1			
1	Ln 14	L14: Maybe refer to “large sharks” as being apex and targeted here in the abstract for sake of consistency with the rest of the manuscript.	This has been updated
2	Ln 33	L33: I believe “if” should be “is”.	This has been updated
3	Ln 59	L59: I assume it’s meant to be “sharks play a key role...”	This has been updated
4	Ln 122	L121: “even though species was likely they have been caught previously” is a bit strangely written; I’m assuming this is meant to be something like “species was likely to have been previously caught”.	This has been updated
5	Ln 166	L168: I’d like to see brief definitions given to functional uniqueness and functional specialisation, as done for functional richness earlier, so that their full context is clear to a reader. This may grant opportunity for extra discussion about the susceptibility of the assemblage to further losses too.	This sentence has now been updated “Functional uniqueness (which identifies the overall isolation of a species and is linked to functional redundancy) and specialisation (which identifies species that contribute the most to functional richness) metrics are suitable for assessing the levels of redundancy in the community and whether species in the community are more generalist or specialist in nature.”
6	Ln 376	L374: Please state which function was used in the mFD package to calculate functional uniqueness and specialisation. I’m presuming it’s the “fuse”	This sentence has been updated to reflect this comment

		function, but a reader may not necessarily have experience with this package. Thanks.	"We calculated the species specific metrics functional uniqueness (FUn) and functional specialisation (FSp) using the fuse function in the mfd package "
--	--	---	---